# Digital Mental Health Interventions for Young People in Rural South Africa: Prospects and Challenges for Implementation

**DOI:** 10.3390/ijerph20021453

**Published:** 2023-01-13

**Authors:** Tafadzwa Mindu, Innocent Tinashe Mutero, Winnie Baphumelele Ngcobo, Rosemary Musesengwa, Moses John Chimbari

**Affiliations:** 1Department of Nursing and Public Health, University of KwaZulu-Natal, Durban 4041, South Africa; 2Department of Psychiatry, University of Oxford, Oxford OX3 7JX, UK; 3Research and Innovation, Great Zimbabwe University, Masvingo P.O. Box 1235, Zimbabwe

**Keywords:** adolescents, apps, digital, mental health, rural, social media, South Africa, young people

## Abstract

Globally, most young people living with mental health conditions lack access to mental health care but have access to a mobile device. The growing access to mobile devices in South Africa has the potential to increase access to mental health care services through digital platforms. However, uptake of digital mental health interventions may be hampered by several factors, such as privacy, confidentiality, informed consent, and affordability. This study identified the prospects and challenges of implementing a mobile phone-based mental health intervention for young people in Ingwavuma area. Data were collected from 93 young people in three villages purposefully selected in Ingwavuma area. Participants included in the study were aged 16–24. Data were collected through a questionnaire. Thematic and descriptive analysis was performed on the qualitative and quantitative data, respectively. Mental health education was low, with only 22% of participants having received prior education on mental health. About 50% of the participants had come across a mental health app, but none of them had used any of these apps; 87% of participants had Internet access; 60% preferred to use social media to contact a health worker; and 92% suggested that use of digital apps would improve mental health literacy among young people. Barriers to access of digital mental health interventions were identified as the high cost of data, restrictive religious beliefs, limited privacy, lack of native languages on most digital platforms, low digital literacy, and complicated user interface. In uMkhanyakude, uptake of digital mental health apps among the young people was low. We recommend that, developers create context-specific digital applications catered for young people from different cultural backgrounds. Socio-economic issues such as affordability also need to be addressed in developing these tools.

## 1. Introduction

Adolescent mental health is a key public health concern globally [1]. Policy development and implementation for child and adolescent mental health programs is slow and haphazard, particularly in low- and middle-income countries, yet young people represent up to 50% of these populations [2]. The WHO defines mental health as “a condition of well-being in which the person understands his or her own abilities, can cope with the usual demands of life, can work successfully and fruitfully, and is able to contribute to his or her community” [3]. In South Africa, young people are at risk of suffering from mental illness and common mental health disorders due to being exposed to high rates of HIV infection, substance use, and violence [4,5,6]. Generally, young people are defined as those between the ages of 10 and 24, while adolescents are those between the ages of 10 and 19 years, and youth are those between the ages of 15 and 24 [7].

Approximately one in every six South Africans may develop a common mental disorder (depression, anxiety, or drug use disorder) [8]. The South Africa Stress and Health survey revealed a 16.5% 12-month prevalence of common mental disorders (CMDs) which included anxiety, mood, impulse control and substance use disorders [9]. Among the mental illnesses known as anxiety disorders there is generalized anxiety disorder (GAD), panic disorder, phobias, social anxiety disorder, obsessive-compulsive disorder (OCD), and post-traumatic stress disorder (PTSD) [10]. Depression disorders can also be sub-categorized into depressive disorders (major depressive disorder, or depressive episode) and dysthymia (a chronic or persistent form of mild depression); the symptoms of dysthymia are similar to those of a depressive episode but tend to be less intense–these include low mood, loss of interest and enjoyment, and decreased energy [10]. Depression among young people is important to address because it prevents the accomplishment of crucial developmental tasks (such as academic success, navigating changes in family relationships, and forming peer networks) [11].

The COVID-19 pandemic led many health policy makers to recognize the need for accessible mental health support in rural and urban areas [12]. Albeit disastrous, the pandemic also led to the increased use, and adoption of digital health services [13]. Currently, South Africa has a shortage of mental health care workers and a low uptake of mental health services. The use of mobile phones and social media platforms can provide a solution in closing the gap between health facilities and patients needing mental health support. Such digital technology has the potential to change the way mental health illnesses are discovered, treated, and prevented [14]. During the COVID 19 pandemic, South Africa employed HealthAlert, a WhatsApp-based hotline that used automated response and triage to disseminate correct and timely COVID-19-related information from the National Department of Health to the public [15]. MomConnect (another WhatsApp based health intervention in South Africa) has achieved unparalleled coverage of pregnant women by enrolling over 1.5 million pregnant women, accounting for about 60% of this target group over three years [16]. Another mobile based innovation released in South Africa during the COVID 19 period was the MalariaConnect mobile application which has been used to increase malaria reporting, monitoring, and speed up malaria notifications during COVID-19 [13].

Such innovations demonstrate the potential that mobile phones, social media platforms and apps have in improving the delivery of mental health interventions targeting young people. Digital innovations can be used for mental health education, self-diagnosis, decision support system, after clinical care, treatment monitoring, drug adherence and managing substance abuse [17,18]. South Africa is classified as a middle-income economy. However, its health care provision is plagued by inequities across socioeconomic status, race and gender; presenting ongoing challenges, particularly in rural areas [19]. Rural communities are among the most disadvantaged in terms of accessing quality healthcare, with consequent poor health outcomes [20]. Rural communities also face a range of challenges associated with digital accessibility and connectivity which apply in both the physical and virtual sphere [21]. They also contend with barriers to accessing e-health services, such as the cost of data or subscription fees to access digital health apps or platforms on social media.

This study contributed to a larger study conducted in five African countries titled the Ethics of Digital Mental Health Innovations for Young People in Africa (EMDIYA). The aim of the EMIDIYA network was to make recommendations for the design of an ethical framework for the development of digital mental health (DMH) interventions for young people in Africa. Our aim for this study was to identify the prospects and challenges of implementing a mobile phone based mental health intervention for young people in Ingwavuma area. We, specifically, assessed mental health literacy, access to mobile devices and the Internet, awareness of digital mental health interventions, user preferences for digital mental health interventions and barriers to access of digital platforms among young people in the study area.

## 2. Materials and Methods

### 2.1. Study Area

The study was conducted in Ingwavuma Community, uMkhanyakude District in KwaZulu Natal, South Africa in the year 2021 in the month of August. The area has a small town located along the border of Ndumo Reserve [22] and has several villages including Ndumo, Makhana and Bhambanana. Ingwavuma community is characterized by low income and poor infrastructure. 20% of the people are employed and 70% are not active economically [23]. Most of the population in the district have attained low formal education, 65% primary, 57.3% secondary, 3.5% attained tertiary qualification with 32% having no schooling at all (Statistics South Africa. 2016).

### 2.2. Population and Sampling

We sampled 150 participants from three villages namely: Ndumo, Makhane and Bhambanana. We used random sampling to select our study population. At the first stage we selected the three villages purposefully for two reasons: (i) the existence of vibrant young people’s associations, and (ii) the existing relationship between the researchers and the researched community which dates back to 2014, hence selecting these three villages was for continuity and sustainability. At the second stage we identified our target population, which was youths aged between 16 and 24 years.

We then determined that in each village we could obtain at least 50 participants using systematic random sampling. Our recruitment was done by community research assistants (CRAs). The CRAs moved into the villages starting from the middle of the village and going in an outward direction counting every fourth house and then entering to recruit. Where there was no young person they went to the next immediate household, and only skipped four when they found a willing participant.

### 2.3. Recruitment Criteria

Participants were included in the study based on age criteria (16–24), this age group was selected based on the specified definition of participants by the funder, that youths are those aged from 14–24. We however chose to recruit from age 16 to 24 years because this age was easy to access from their homes since most of them were matriculants (they had completed high school). Our participants were recruited randomly through door-to-door household visits in the three villages selected for the study. We approached participants in areas where we had already worked in our previous projects so that we could take advantage of the familiarity we had within the village and with the village heads. With the help of CRAs who were familiar with the community, participants were visited in their households and introduced to the program. The CRAs then went on to request permission to administer the questionnaire to any youth in the household who was willing to participate.

### 2.4. Study Design

The study was a mixed-methods study conducted using a questionnaire tool. The questionnaire had mixed questions (close ended and open ended). It was administered by trained CRAs. The questionnaire tool was written in isiZulu language and data were collected digitally using Kobo Collect, an online mobile data collection tool [24]. The questionnaire included multi choice questions, open ended questions and dichotomous questions. We chose the mixed methods study design because the nature of our study was such that we needed to find out what the youths knew about digital mental health and what their preferences and reservations were towards such interventions especially from an ethics point of view.

Open-ended questions helped us to obtain deeper explanations for certain behavior or preferences without necessarily putting any options for the respondent. We were better able to ask questions which required them to report their knowledge such as “what is your understanding of mental health” or “what type of digital platform have you used to access mental health services.” Such open-ended questions helped us to determine the actual extent of knowledge, variations in access, typical preferences, personal challenges etc. Close-ended questions helped us to simple measure and generate a count of the issues that were pertinent to answering the question of access and extent of use. It is important to note that the mixed method design helped us to strengthen our findings, because what we lacked in terms of numbers, we made up for with the qualitative analysis (this became an advantage given that our sample recruitment yielded a low sample size).

### 2.5. Questionnaire Description

The collected data captured information on mental health knowledge among the adolescents; access to mental health services; access to digital devices, the Internet as well as social media; knowledge of digital mental health interventions; preferences for digital mental health services; and challenges with the use of digital platforms or social media. See Table 1 for a detailed description of the tool. See also the Appendix A with the tool. We asked questions on what they thought mental health was and how prevalent they thought the disease was in their community. This was asked to determine whether they knew what mental health was and if they thought the problem was prevalent in their community. We also asked about access to mental health services in order to find out about the centers for mental health treatment in the community and the preferred places to go for people facing this problem.

We also asked questions on digital devices and social media platforms used by the participants to communicate and interact with health workers, this helped us to determine the extent of access to digital devices in this community. We also determined their preferred platforms, and their level of Internet and social media use so as to understand whether they had access to the Internet and how frequent they used it together with the social media platforms. This information helped us to know which digital based mental health interventions would be easier for the young people to adopt.

We asked participants if they had any access to or knowledge about digital mental health platforms, and if they had used them before, this helped us to determine if there were already any participants who knew about these platforms and determine their familiarity with these platforms. We also asked them about the advantages and disadvantages of the social media platforms they were using and recommendations they had for the promotion of digital mental health interventions on these social media platforms. Lastly, we asked about the barriers and challenges likely to affect use of social media and other digital based mental health platforms. This helped us to identify the factors that could hinder the successes of social media based mental health interventions in this community.

### 2.6. Data Collection

Data were collected over a period of six days, we spent at least two days in one area with CRAs moving into the households following the randomization order we had established. We had three CRAs and they were each place at different points in the village and they moved around households forming three different clusters. Each questionnaire was about 30 min long and the CRAs conducted at least 6 h of work per day. The CRAs administered the questionnaire and entered the data onto the device which had the questionnaire.

### 2.7. Ethical Considerations

All participants took part in the study after being informed about the purpose of the study, what was required of them and their rights as research subjects. For participants who were below the age of 18, assent was obtained from parents or guardians. The survey was administered at participant’s households; hence research assistants could engage the parents of under 18 participants to request permission to involve them in the study. The first step in the data collection was requesting permission to do the study from the village heads, and then we would go on to the household head of the compound. Then, when household head consents to the research we would proceed with requesting those qualifying in our age criteria to participate and complete the consent form.

Younger participants (below 18) were engaged with the consent of the parents. During the data collection processes, researchers first briefed the participants about the purpose of the study. After the briefing, researchers asked all the potential participants to sign if they were willing to participate in the study. The researcher worked closely with community research assistants (CRAs) throughout the process. CRAs were capacitated with data collection skills prior to the data collection processes. The study received ethical approval from the University of KwaZulu-Natal Human and Social Sciences Research Ethics (HSSREC).

### 2.8. Validation of the Tool

We piloted the tool with an initial group of 43 participants recruited during workshops conducted in the three study areas when the project was launched. The responses from this pilot were analyzed and changes to the tool were suggested, hence in this study we present only the outcomes from the 93 participants who were interviewed by CRAs after the validation of the tool.

### 2.9. Data Processing and Analysis

Quantitative data were analyzed using SPSS 27. Data were validated, edited, and coded. Quantitative analysis yielded descriptive data, which we analyzed descriptively. We used mainly percentages to show the difference between the male and female participants. For each question, we showed the total count and percentage of participants who responded in a certain way. Our method of analysis was simple descriptive, we did not conduct any further statistical calculations except to determine in simple the count of responses for each question.

For our qualitative results we used inductive analysis. Participants typed in their responses in IsiZulu and these were translated into English. We looked for common themes within these responses, similar responses worded differently were identified and coded with numbers to indicate that they belonged to a specific category or theme. The collected themes were either quantified and reported as counts or presented qualitatively as key outcomes. In the description of the qualitative results, we put the interesting statements from each identified themes as quotes, to demonstrate the meaning derived we followed up each group of quotes with an explanation of what the responses meant in the context of the study.

## 3. Results

### 3.1. Demographics

We had a total of 93 participants (52 female and 41 males) out of the anticipated 150, 57 participants did not respond to our questionnaire. Participants who responded were aged between 16 and 24. (87.1%) participants had attained a Matric (GSCE/A level) level of education and (11.8%) had primary education only one participant had a college education. Almost all (*n* = 92) of the participants were single. Some had one child (20%, *n* = 19), and a few had two or more children (5%, *n* = 5). See Table 2.

### 3.2. Mental Health Awareness

Only a few participants (22%, *n* = 21), had been taught about mental health, the rest (77.4%, *n* = 72) had not received any prior education on mental health. See Table 3. Those who said they had been taught before, got the information from school, the local clinic, a community health worker and other community health outreach programs conducted in the area. None, had got it from an online or digital source.

### 3.3. Identification of MH Conditions

The most common mental health conditions identified by participants were substance abuse/misuse (60.2%), post-traumatic stress disorder (33%), depression (31%) and anger (13%). See Table 3. The participants associated most mental health problems in the area to unemployment, stress, drugs, and poor upbringing.

When asked to describe what mental health is, participants gave several answers. Most participants associated mental health illness with depression, they mentioned the words stress and overthinking associated with these problems. Some said this comes along with poor appetite and chronic headache. Others associated mental illness with schizophrenia, they mentioned one being mad and beating up people or doing random and odd things such as walking around aimlessly and talking on your own. Some youths associated mental illness with self-isolation and fear of being around others. They described people who keep to themselves and avoid contact with other people by all means. Others mentioned mental problems associated with poor upbringing and physical abuse from households and sexual partners


*“Having stress and losing weight because of not eating.”*
—Male 19


*“It is when you find yourself having a problem and you do not know how you will solve it.”*
—Female 16


*“Sometimes one beats people, and they roam along the road attacking people.”*
—Male 18


*“One does not want to be near people, they want to be alone all the time.”*
—Female 18


*“Having a temper or violence towards family and other members of the community.”*
—Female 20


*“Having a chronic headache.”*
—Male 18


*“Mental problems sometimes come from the fact that we come from different households, some people come from homes with abusive/violent parents.”*
—Female 20


*“I know that the mind will be affected when one uses too much drugs.”*
—Female 22

### 3.4. Availability of Mental Health Treatment in Rural Communities

In order to obtain mental health treatment, participants said people either go to the clinic/hospital (73%), visit a health care worker (25%), use traditional medicine (12.9%), visit a spiritual healer (3.2%) or go to a church for prayers (9.7%). When asked about the type of mental health services that they thought young people could make use of in their area they selected the following: counselling (84%), rehabilitation (16%), and medicinal treatment (19.4%) (see Table 4).

### 3.5. Access to Digital Devices and Internet-Based Mental Health Platforms by Rural Based Youths

All the participants had access to a mobile digital device. The types of devices that the participants used for accessing the Internet were mostly smart phones (91%), only a few had tablets (5%) and computers (2%). Regarding Internet use, most of the participants stated that they were frequent users of the Internet (87%), while others said they did not use it as often (11.8%). Of the participants, 83% said they had the ability of using digital apps while 16.1% (*n* = 15) said they could not use them.

About 49% of the participants in the study had come across a mental health app. However, those that mentioned having come across digital mental health apps failed to adequately name or describe these apps. Table 5 shows the responses on questions about access to digital devices. Of the participants, 92% stated that it was important/relevant to use digital apps to create mental health awareness among young people in their community. 67% (*n* = 63) said they had a responsibility to access digital mental health app, while only 32.3% (*n* = 30) thought it was not necessary. A majority of the participants, 74% (*n* = 69), thought that it was necessary to have a mental health app on one’s phone while 25.8% (*n* = 24) believed it was not necessary.

97.8% (*n* = 91) agreed that they needed to have a social media platform for local young people to discuss mental health issues. 83% (*n* = 78) agreed to join a social media group for discussing mental health issues and gave their cell phone numbers so that they could receive a link to join the WhatsApp group. 60% (*n* = 56) of participants preferred contacting the health worker via social media (Facebook, WhatsApp and Instagram).

### 3.6. Acceptability of Social Media as a Mental Health Intervention Tool for Rural Based Youths

Participants suggested that there was a need to have social media platforms for discussing mental health, however they had limited understanding of the privacy features of social media platforms. They said that the available social media platforms had limited privacy settings which allowed people to see their information and photos. Participants complained about being stalked and followed by strangers and some downloading their pictures and videos. Fears claimed were risks of people accessing personal details of a user and being able to see what they post and downloading pictures of participants.


*“I don’t like it when you use Facebook and meet strangers and fall in love with them not knowing that it is a criminal trying to kill you.”*
—Female 19


*“I don’t like it when people download your videos and use them without consent.”*
—Female 17

Other disadvantages that were mentioned were related to indecency on most of the social media -based platforms. They also mentioned verbal attacks on social media, frequent use of foul language by others, mocking, criticism and spreading of lies and misinformation as some of the major challenges they face while using social media platforms. Another concern was that women were being demeaned on social media due to sharing of indecent pictures. There were some comments related to identity crisis and issues of culture/lifestyle disparities which could make some uncomfortable on social media platforms


*“The proliferation of indecency on social media is a barrier for some.”*
—Female 18


*“People are affected by low self-esteem as a result of comparing themselves to others.”*
—Male 20

As for the advantages, the participants mentioned that platforms such as Facebook had a free mode which allowed them to see text-based posts and also send/receive messages from friends and relatives. Other significant advantages identified were: the ability to communicate easily with people who are far, the ability to discover information and learn, and that the platform was intuitive.


*“What I love with social media is that you can get any information or help that you need, for example you can get assistance with a school assignment.”*
—Male 19


*“Meeting and interacting with people from distant places.”*
—Male 19

These comments by the participants point to some of the limitations that can make an intervention that is based on social media platforms hinder participation. Some of the reservations expressed by the participants indicate the possibility of young people feeling more awkward as they interact with others in group settings. However, these issues can be managed through use of chat-based interventions where its one on one with youths or sharing of mental health literacy material via the inbox of participants instead of in the public posts.

### 3.7. Structural Factors to Consider in the Development of DMH Platforms for Young People in Rural Areas

***a.*** ***Easy user interface:*** Participants mentioned that some young people may not be tech-savvy, hence the apps needed to be made user friendly to accommodate the illiterate.


*“That they should make it (Apps) easy (simple) so that we can all be able to use.”*
—Male 19


*“They should understand that not everyone has good knowledge about apps.”*
—Female 22

***b.*** ***Literacy and language:*** They also mentioned that most people have reservations due to the language used on most platforms, they feel left out. They asked for developers to adapt apps to their local languages.


*“That we are not English speakers.”*
—Female 20


*“They should use all (local) languages so that all of us can understand.”*
—Male 19


*“Use of English on most platforms make it hard for us to use the apps.”*
—Male 20

***c.*** ***Cost of data:*** Participants were concerned about the cost of data and needed apps that are subsidized, free or that work offline.


*“There should be free WIFI at our schools.”*
—Female 17


*“That the youths do not work and hence they have no money to buy data.”*
—Female 19


*“They should make apps with free mode like Facebook.”*
—Female 19

***d.*** ***Consultation and engagement:*** The participants mentioned that developers can come to their areas to consult with them and interact with them in person so as to understand their issues in context. Apart from just making things available on apps they needed to be visited and educated about these platforms and also about mental health.


*“The other thing is that they should come to the communities where we live.”*
—Female 18


*“That they should come to our community and educate us because some of us do not have personal phones.”*
—Female 18


*“That they should come to our community and engage us.”*
—Male 18

***e.*** ***Religious and cultural beliefs:*** participants mentioned that most religious and cultural beliefs were against the use of social media.


*“Some of our religious congregations are against the use of social media.”*
—Female 18


*“Religious beliefs against use of social media.”*
—Male 18


*“Traditional healers predominant.”*
—Female 22

## 4. Discussion

The aim of our study was to identify the prospects and challenges of implementing a mobile phone based mental health intervention for young people in Ingwavuma area. Our assessment revealed that the level of mental health literacy was inadequate. We also found that access to mobile devices and the Internet was good, however, awareness and use of digital mental health interventions was poor. Participants supported the idea of implementing digital mental health interventions using social media platforms. The mental health interventions we considered to implement via this platform are those promoting psychoeducation, this is something that can help to improve mental health literacy, as well as enable youths to find coping strategies when suffering from mental illness. The barriers to access among young people in the study area were identified and grouped under prohibitive religious beliefs, poor literacy, lack of privacy etc. These barriers can be overcome by making sure that the views of the young people are incorporated into the programming and design of interventions. There is a need to bring solutions that are contextual to the rural setting and applicable to the social, economic, cultural and religious backgrounds of the youths.

### 4.1. Mental Health Access and Awareness

Our findings showed that participants did not know much about mental health. They had limited access to mental health interventions and were not very familiar with any digital mental health platforms. Other studies in South Africa report similar outcomes, with one study reporting that 65% of young people in South Africa face some form of mental health challenge, and they are unable to seek help [25]. In sub-Saharan Africa the treatment gap for people with mental, neurological and substance abuse disorders is vast with only 10% of these people receiving treatment [26]. We found that there was a need for increasing mental health literacy for young people (preferably) using accessible online platforms such as social media. Young people could benefit a lot if they were taught about mental health on their social media platforms.

Mental health literacy is an important component of health literacy and is essential for improving access to mental health care and reducing stigma related to mental illness [27]. Without awareness or mental health literacy, many young people will continue to suffer untreated mental health ailments. It is desirable that any health interventions meant for young people be tied to the way they access digital health information [28]. This necessitates the establishment of appropriate digital channels for reaching out to the young people. Developers, researchers, and health practitioners should find ethical ways to use digital mental health interventions to serve marginalized communities [29].

### 4.2. Prospects for Digital Mental Health Interventions

Our study found that participants had no experience of using mental health applications of any nature. About half of them knew about apps for mental health support, but none could describe any that they had used before, indicating a lack of experience using any of those apps. We found that participants were more accustomed to social media platforms and that integrating mental health services into these platforms would be an easier and accessible approach to developing a digital based mental health intervention. The potential to use and customize social media based mental health interventions for prevention and treatment purposes is high [30]. Our participants suggested common social media platforms such as Facebook and WhatsApp to be used to reach out to young people in order to increase mental health literacy and provide support.

The use of social media among people living with mental health disorders is noticeably growing, with most patients using these platforms to share their lived experiences with mental health illness and to also get help from professionals [31]. Online platforms such as Facebook are both convenient and cost-effective since they have potential to reach many numbers [32]. One study conducted in Canada evaluated the effectiveness of a brief social media intervention for mental health among youths; they found that one year later, there were noticeable improvements in young people’s views regarding mental health issues. The campaign, however, struggled to inspire young people to adopt healthy mental health practices and to provide them the resources they needed to feel capable of assisting someone who may be struggling with mental health concerns [33]. The reason for these limitations may be due to the briefness of their interventions.

Another study evaluating the impact of social media interventions found that participants benefited from interacting with peers online through sharing personal stories and coping mechanisms for dealing with the day-to-day problems of living with a mental illness [34]. In another study reporting on mental health sufferers from 10 countries, the majority of participants indicated interest in social media-based mental health programs, particularly those that promoted general health and wellbeing and helped people cope with mental health symptoms [35]. Another study conducted in Nigeria which evaluated the efficacy of a social media-based intervention in lowering youth drug abuse tendency reported that the WhatsApp based intervention they used effectively reduced drug abuse propensity for participants in the treatment group compared with those in the control group. Two years later a steady decline in drug use propensity was observed in the treatment group [36]. These findings showcase the ability of social media to be effectively used in the Ingwavuma setting, however, with certain considerations which we discuss below.

### 4.3. Challenges of Using Digital Health Interventions among Young People

Our study participants reported that privacy, security, and confidentiality were a concern on social media platforms, and this could pose as an ethical barrier to the implementation of digital interventions. There is a growing urge to work with young people to mitigate the concerns raised about privacy risks associated with using social media as a mental health intervention [37]. Many rural communities in developing countries are not experienced to the ethical concerns related to privacy on digital platforms, and the legal consequences of exchanging information on these platforms [38].

Another issue of concern from participants was that of affordability, there were reported challenges with accessing data due to the high cost of airtime and it was recommended that when developing and implementing digital mental health innovations, developers should ensure that their innovations are free to access. A study on ICT access in South Africa affirmed that cost and affordability of digital mobile devices and services was a huge barrier to Internet use by South Africans [39]. The high costs of sophisticated apps and services limits the potential transformational benefits of ICT [40]. In South Africa few households (10.4%) have access to the Internet at home [41].

According to our study there was Internet reach in the study area. However, the Internet reach did not translate to effective access as it was limited to exorbitantly priced WhatsApp and Facebook bundles provided by broadband service providers. The results indicate that the costs of data were too high for the young people to afford. It is true that they have access to an Internet connection but the ability to purchase data and connect is a challenge. Infrastructure-wise, rural areas such as Ingwavuma rely on mobile broadband operators whose rates are very high. There are no cable-based options for Internet connectivity, hence the inability to connect.

Participants also mentioned the lack of cultural relevance in most of the apps. This led to limited use of apps by many whose religious beliefs discouraged the use of social media and digital devices. Limited locally relevant content and failure to see the relevance or usefulness of digital technology is a barrier to use of apps by local people [40]. Such ethical barriers can be avoided by ensuring that apps are representative of the communities they serve. Uptake of DMH innovations can be facilitated by the development of specific design elements, such as program interfaces or content that accommodates or incorporates local values, cultural beliefs, and practices [17].

### 4.4. Recommendations for Future Research

We did not come across any apps, or IT based interventions produced in South Africa meant to be used directly by young people. This is a challenge that needs to be addressed, because we realize that most mental health apps in the market are not tailored for African young people. There is little evidence on the effectiveness of DMH interventions especially in African rural settings, and among young people in particular. There is a need for research that can evaluate the effectiveness of DMH interventions, assess their costs, and analyze their potential risks especially in low resource settings [17]. This research can also help to show how these innovations can bring measurable improvements in mental health outcomes [14].

### 4.5. Limitations

The limitation of this study is in the randomized sampling approach which we used in this study, this is further affected by the low number of participants who responded to the questionnaire. This introduced a disparity in the representation of the adolescents in the community, hence the findings cannot really be generalized. The use of the randomization technique without fully applying a finite sampling method affected the way that these findings were interpreted. Our findings cannot be generalized, and they need to be supported by evidence from similar studies in the same area or similar context, however, such articles are few.

The low participant outcome was a result of community research fatigue, our CRAs failed to access the desired number of households in the village because several households had been visited more than twice during earlier studies conducted under a different funder. The household heads no longer wanted to respond to questionnaires, but they were now expecting hand-outs or to hear about any benefits accruing from study outcomes. Hence CRAs were denied entry into households if there was no benefit for the households. Despite this limitation, we consider that the findings do offer some credibility given that the margin of similarity in the responses of participants was high.

## 5. Conclusions

The growing access to mobile technology by rural communities presents opportunities for implementing digital mental health innovations. Our study found that knowledge and use of such innovations among young people was minimal in the Ingwavuma area. Despite this, there was a promise that uptake of digital mental health interventions was possible considering that young people had access to digital platforms and were willing to be engaged in the delivery of mental health interventions through social media. There is a need to make these social media-based interventions culturally relevant, free to access, private and user-friendly in order to increase use by young people. These findings are important to the public health sector in South Africa as they help provide information about the use of digital mental health interventions by youths. Post the COVID 19 era, many people and health institutions are considering investment in more web-based solutions for helping patients, and it is therefore key to understand what works for young people and what does not. This study provided this information in the context of a rural setting.

## Figures and Tables

**Table 1 ijerph-20-01453-t001:** Showing the main questions that constituted the questionnaire used for the study.

Category	Question
Demographics	Marital status
	Education level
	No of children
Mental health awareness	Ever been taught about mental health
	Common mental health problems affecting youths in your community?
	Where do youths seek treatment or assistance when they have mental health problems?
	What type of mental health services do youths need?
Digital interventions for Mental Health	Do you know about digital mental health apps
	Have you ever used any digital mental health/digital health app
YP’s access to digital devices & Internet	What digital devices do you use to access the Internet
	Which social media platforms do you frequently access
	How frequent do you use the Internet & social media platforms
YP’s preferences for digital mental health interventions	What are you preferred methods for contacting a health worker
	What are you preferred platforms for receiving mental health advice
	Do you recommend youths to use mental health treatment apps
Barriers to use of digital mental health innovations	Do you have knowledge or the ability to use digital apps
	What ethical issues should be considered in developing MH apps for youths in your community

**Table 2 ijerph-20-01453-t002:** Showing the demographic characteristics of the study participants.

Question	Response	Total		Male		Female	
		*n* (93)	%	*n* (52)	56%	*n* (41)	44%
Age	16–18	26	28%				
	19–21	41	44%				
	22–24	25	27%				
Marital status	Married	1	1%	0	0%	1	1%
	Single	92	99%	41	44%	51	55%
No of children	1 child	19	20%	7	8%	12	13%
	2 children	4	4%	1	1%	3	3%
	3 children	1	1%	1	1%	0	0%
Education level	Matric	81	87%	32	34%	49	53%
	Primary school	11	12%	8	9%	3	3%
	College	1	1%	1	1%	0	0%

**Table 3 ijerph-20-01453-t003:** Showing the level of mental health awareness, common mental illnesses and the available avenues for treatment.

Question	Response	Total		Male		Female	
		*n* (93)	%	*n* (52)	%	*n* (41)	%
Ever been taught about mental health	Yes	21	23%	9	10%	12	13%
Common mental health problems affecting youths in your community?	Substance Abuse	56	60%	26	28%	30	32%
	Depression	29	31%	12	13%	15	16%
	Anger	12	13%	5	5%	7	8%
	Post traumatic disorder	31	33%	16	17%	13	14%

**Table 4 ijerph-20-01453-t004:** Showing responses of participants to the question on availability and type of mental health services.

Question	Response	Total		Male		Female	
		*n* (93)	%	*n* (52)	%	*n* (41)	%
Where do youths seek treatment or assistance when they have mental health problems?	Mental clinic	68	73%	31	33%	37	40%
	Community health worker	24	26%	13	14%	11	12%
	Sangoma	12	13%	5	5%	7	8%
	Spiritual healer	3	3%	1	1%	2	2%
	Church	9	10%	3	3%	6	6%
Type of mental health services do youths need?	Counselling	79	85%	35	38%	44	47%
	Rehab	15	16%	6	6%	9	10%
	Medicinal Treatment	18	19%	7	8%	11	12%

**Table 5 ijerph-20-01453-t005:** Showing the outcomes to questions relating to access to digital platforms and the.

Question	Response	Total		Male		Female	
		*n* (93)	%	*n* (52)	%	*n* (41)	%
Awareness of mental health apps	I have heard about mental health apps	46	49%	9	10%	12	13%
	I have used a mental health app	0	0%	0	0%	0	0%
	Mental health apps are important	92	99%	40	43%	52	56%
	Mental health apps can benefit youth	92	99%	41	44%	51	55%
Digital devices used to access Internet	Phone	85	91%	39	42%	46	49%
	Tablet	5	5%	0	0%	5	5%
	Laptop	2	2%	2	2%	0	0%
	Computer	2	2%	1	1%	1	1%
Social media platforms accessed by young people	WhatsApp	30	32%	10	11%	20	22%
	Facebook	60	65%	31	33%	29	31%
	Instagram	4	4%	2	2%	2	2%
	Twitter	6	6%	3	3%	3	3%
Frequency of using digital platforms	All the time	81	87%	36	39%	45	48%
	Not very often	11	12%	5	5%	6	6%
Preferred methods for contacting a health worker	Mobile phone (App, text, calls)	37	40%	18	19%	19	20%
	Social Media (FB,WhatsApp, etc)	56	60%	25	27%	31	33%
	Website	17	18%	8	9%	9	10%
Preferred channel for receiving mental health information	Website	10	11%	7	8%	3	3%
	Mobile App	12	13%	6	6%	6	6%
	Social media	34	37%	12	13%	22	24%
	Voice or SMS	37	40%	18	19%	19	20%
	Video media	20	22%	11	12%	9	10%
	Audio media	9	10%	1	1%	8	9%
I think youths must use mental health apps	Yes	63	68%	27	29%	36	39%
I believe mental health Apps are necessary for youths	Yes	69	74%	27	29%	42	45%
I have knowledge or ability to use digital apps	Yes	78	84%	35	38%	43	46%

## Data Availability

Data has been supplied as Appendix A.

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
