# Peer review of "Digital Mental Health Interventions for Young People in Rural South Africa: Prospects and Challenges for Implementation"

_ijerph, 2023, doi:10.3390/ijerph20021453_

Round 1
Reviewer 1 Report
Review Report
Brief summary: This is an interesting study that tackles the topic of digital health interventions and their use in the population of adolescents and young adults in South Africa (Ingwavuma Community). It aims to explored factors that might encourage or prevent young people from fully utilising available DMH interventions. A mixed-methods study design was used by the mean of a questionnaire that was conducted with 93 people.
It addresses an important topic on mental health services, especially in rural areas of middle-income countries such as South Africa and is therefore pertinent to the field. Please find below my comments.
Introduction :
- The information conveyed in the introduction is pertinent to this study. It is however suggested to restructure the introduction to account for clarity, for instance, here is the suggestion:
o 1st paragraph: The situation relating to mental health issues in South Africa. In this paragraph, it is expected to define mental health, to define what is meant by ‘’young people’’ as this may vary from a country to another and to highlight the main major mental health issues in South Africa.
o 2nd paragraph: The situation relating to digital health innovations in South Africa. It is suggested to bring the focus on a less broad topic. For instance, the focus could be on mobile applications since the reference to apps or social media use within apps is discussed throughout the manuscript. Digital health is a very broad field.
o 3rd paragraph: The relationship between young people mental health issues in South Africa and the use of mobile applications (or other technologies that you would have defined in the scope of the 2nd paragraph).
o 4th paragraph: Clearly state the aim of the study and your hypothesis. Is this the current aim : ‘’ We explored factors that might encourage or prevent young people from fully utilising available DMH interventions.’’? If so, it is suggested to have a more specific aim as the current one is broader than what has been conducted and discussed in the methods section.
- Line 62: a reference is missing
- Line 71: there is a floating reference (19).
- Line 80-81 : The use of reference 22 is misleading. The paper referenced states that the voices of young people in rural areas who are not in education, employment and training are marginalised: not solely the voice of young people in rural areas. This statement should not be overgeneralized. It is suggested to add this specification.
Material and Methods:
- Line 91-105: It is suggested to remove these sentences as they are part of the template provided by MDPI to structure your section.
- Line 121: It is suggested to remove the two dots ‘’ .. ‘’
- Study population: The following elements should be addressed:
o What is meant by young people? Why were people below 16 of age and above 24 of age were excluded?
o How were the participants recruited?
o 136 participants out of how many potential participants? Why was 136 selected as the sample size?
- Design: The following elements are missing:
o Questionnaire tool: which one was used? (line 125) and why? This is addressed later on (lines 136-137). It should be in the introduction of the design paragraph.
o It is suggested to add the questionnaire as a supplementary material.
o Per definition, the used of a questionnaire with a mixed-methodology is not purely explorative as we are limiting the scope of answers a participant can select (especially with multiple choices questions). The design here is a mixed-method study (considering there is a qualitative and a quantitative component). It would be interesting for the readers that the author state why they selected this methodology and why is it pertinent to address the aim of their study.
o It is suggested to add a Table to highlight the different components of the questionnaire and their target.
o The data collected is vaguely reported from line 130-135. It is suggested to clearly state which data is being collected and why.
- Line 139-145 do not belong to Ethical considerations: this belongs to patient recruitment.
- Lines 144-148 are a repetition of lines 139-145. It is suggested to remove them.
- Please review line 155-156. ‘’The when household (…) ‘’: it is suggested to restructure to account for clarity.
- It is unclear why the research assistant needs to obtain the parent’s permission of the participants that are older than 18 years old. The current paragraph on Ethical consideration seem to hint that all participants must have received prior approval from their legal tutor.
- Who are the ‘’Principal Investigators’’ ? Which team is involved with the project? It is suggested to avoid using broad terms and either described them or you could also use their initials if they are the authors of this manuscript.
- Line 167 : excel should read Excel
Results:
- Demographics: It is suggested to add a Table to highlight the principal characteristics of the participants. Considering the questionnaire was addressed to 136 people and only 93 replied: what happened to the other 43? Did they refuse to participate? Did this change the participants characteristics significantly? It is suggested to report the reasons of the mismatch in numbers of participants and report the dropout rate.
- Mental health identification: The reporting of mental health issues is unclear. What constitute a mental health condition in your manuscript? Please use a validated nomenclature to report mental health conditions: either DSM-5, ICM, or other. Otherwise ‘’Stress’’ is very broad and little can be derived from it. The same applies to ‘’Violence’’ which is not a mental health condition as per global diagnostic manuals such as DSM-5 or ICM.
- Lines 244 to 284 are very difficult to read because of the formatting. It is suggested to restructure these paragraphs to clearly outline: the verbatims from the participants and the idea conveyed.
- Was section 3.6 the only element that contained open-ended questions? If not, verbatims should be added to the other section considering the mixed-methods approach.
Discussion:
- The first paragraph of the discussion should restate the aim of the study and give synthetized overview of the main results.
- Line 295-296: ‘’We found that people could benefit a lot (…) ‘’ : It is suggested to rephrase or review this sentence considering the results were not about the effect of the digital interventions, therefore it cannot be stated that these were findings of the presented study as this is not part of the results.
- The wording mental health intervention is used across this section and the manuscript and never defined. What is meant by intervention: psychoeducation? Treatment-based apps? It is suggested to define this in the introduction and specify clearly what type of intervention is being referred to in the discussion section of the manuscript.
- Please state the limitations of your study.
- Line 390-391: These line state a finding on feasibility: this is not a feasibility study; therefore, no such data can be reported.
- In the conclusion, it is suggested to state why these findings are important and how they might impact future decisions or health directions.
- In the introduction section, it is stated that South Africa as a wide problem with accessibility in terms of digital accessibility and connectivity, yet all the participants had a cellphone and connection to the internet. It is suggested to discuss this as it might introduce a bias in the interpretation of the results.
Minor comments:
- Line 25-27 (Abstract) : there seems to be several typos linked to the formatting. For example, ‘’.-.’’
- Please review the repetition of ‘’the young’’ or ‘’young people’’ across the manuscript. If the population is clearly targeting young people aged from 16-24, it is not needed to repeat this across the manuscript.
Author Response
Find attached

Reviewer 2 Report
The paper is in general moderately written; there are some major concerns, especially in presenting the results, in the description of the research procedure and in the discussion section.
1. Affiliations of the authors should be provided in more detail and correctly.
2. There are a lot of typos in the paper (lines 15, 25-26, 31 71 and in other places). This should be edited very carefully.
3. Please sort the keywords alphabetically.
4. The abbreviations were used incorrectly. Check all the abbreviations in the whole paper, and use them correctly.
5. Lines 91-105? What is it? This must be deleted from the paper. Prepare your paper carefully before reviewing. It is inappropriately to submit the paper with editing errors.
6. When was the study carried out?
7. Lines 139-143 and 144-148? You have the same lines twice. Please reread your paper clearly and fix all editing errors.
8. Lines 167. Please write Excel with a capital letter.
9. The methods, participants and procedure section are poor and have no relevant information. Please describe clearly the participants and their demographics characteristics, procedure (e.g., time period), how you analyzed the data, your methods etc. In the present form, your description is insufficient. This is a very major methodological concern. See this article as an example how to present data clearly and precisely (https://www.mdpi.com/1660-4601/19/22/14894).
10. See also Supplementary materials with the qualitative data (https://www.mdpi.com/1660-4601/19/22/14894). Please prepare this material for your paper.
11. Present all your quantitative data in the form of a table.
12. When you quote statements, please use quotation marks.
13. Please divide your extremely long paragraph into more paragraphs. Long paragraphs are unwanted.
14. Could you provide gender differences and/or other demographics differences for your study variables? This will make your paper more comprehensive and useful.
General discussion.
15. The weakest part of the paper is the results section. I recommend presenting the results in the form of a table. The discussion section should be referred more to the results of the study. Please provide sociodemographic comparisons.
I believe if you make these corrections, your paper will be good and may be published. I am waiting for your revision. Please address my concerns carefully.
Have a nice day!
Author Response
Find attached

Reviewer 3 Report
Dear authors,
I am sure that digital psychological help (as wekk as the barriers associated with it) is is a very important problem. This is especially important for countries and regions where access to face-to-face psychological support is difficult. SoI think your manuscript should be published. However, I have a few questions:
1) What the questionnaire included (I'm sure it should be described in more detail)
2) Why does section 2 talk about 136 participants, and section 3 about 93?
3) It seems important to me to understand which participants have successfully used digital psychological support? I think it might be useful to describe a group of young people who are more successful in using the resources that are available.
Please note: it seems that there was technical information in the text of section 2. The list of references should be drawn up more carefully .
Round 2
Reviewer 1 Report
This is the Round 2 of reviews for this interesting study that tackles the topic of digital health interventions and
their use in the population of adolescents and young adults in South Africa (Ingwavuma Community). The authors have conducted modifications according to the previous comments received.
Please see below my additional comments:
Introduction:
- Lines 42-43 should not be in the introduction.
- Lines 44-47: while there has been an attempt to add common mental disorders, the reported data comes from an outdated study from 2008 (Williams DR, Herman A, Stein DJ, et al. Twelve-month mental disorders in South Africa: prevalence, service use and demographic correlates in the population-based South African Stress and Health Study. Psychol Med 2008;38(2):211-220.) That was referred to in the reference provided by the authors (reference number 9). ‘’ depression, anxiety, or drug use disorder’’ are categories of common disorders. A deeper understanding of the mental health problematics (i.e: major depressive disorder, adjustment disorder, generalized anxiety disorders, etc.) should be made explicit. Otherwise, it is too broad to simply state that ‘’South Africans are suffering from anxiety’’ as this is too common.
- Lines 39-40 : The authors state that in South Africa ‘’(…) young people are more at risk of suffering from mental illness and common mental health disorders due to being exposed to high rates of HIV infection, substance use, and violence (…)’’. They are more at risk as compared to whom? The reference provided, for example, reference #6 is not identifiable (please see the list of reference as it seems to be miss-referenced).
- In the second paragraph, the authors added examples of innovations and mobile apps used in South Africa. This is excellent as it gives a better insight as to what has been done and what can be done as well as the impact on the community.
- Line 78 ‘’ It is against this background that we assessed (…) ‘’ could be replaced simply by ‘’We assessed (…)’’ considering it is implicit that the study aims comes from the previous background.
- Lines 83 to 87 coming after the implicit statement of the aim of the study is confusing for the readership as it is not clear what aims is being tackled. It is suggested to bring these lines prior to the statement of the aims as your study is part of what the EMIDIYA is attempting to achieve.
Materials and Methods:
- Lines 90-92 : the statement about the research team working in this facility since 2014 is not relevant to the presented study.
- Lines 113-114: This needs to be re-structured. From what is understood:
o You identified 150 respondents (50 per village) : this is part of the design
o 93 people responded to your questionnaire: this is part of the results
o 57 people did not respond to your questionnaire: this is also part of the results
o Lines 114-120 is the explanation for the 57 people who did not respond to your questionnaire: this is part of the discussion (limitations).
- Lines 125: school leavers is unclear.
- The random selection process limits greatly the interpretation of the study considering it appears that no finite sampling was conducted ‘’participants were recruited randomly through door to door household visits in the three villages selected for the study’’. This should be discussed in the limitations of the study as to how it impacts the results.
- Lines 213-216 : this is a result.
Results:
- Table 2 should be included in section 3.1
- Is sex discrepancy important in the present study. If not, Male/Female in Table 2 is a result and should not be part of column titles.
- The reporting improved in this version of this manuscript.
Discussion:
- The aims reported are mismatching :
o Introduction ‘’ we assessed the prospects and challenges of a mobile phone based mental health intervention for young people in Ingwavuma area’’
o Discussion ‘’ identify the prospects and challenges of implementing a mobile phone based mental health intervention for young people in Ingwavuma area’’
o Assessment and identification are two different concepts. Your manuscript focuses on the identification of prospects and challenges. There is no assessment being conducted.
- Section 4.5 should add the limitations of the random selection design that was selected and the impact on the generalizability of the results.
Reviewer 2 Report
The revised version is in general good. Some typos should be fixed.
See for example line 524 (incorrect references): [15][10].
Author Response
The revised version is in general good. Some typos should be fixed.
See for example line 524 (incorrect references): [15][10].
The typos have been addressed.